# Message-passing Selection: Towards Interpretable GNNs for Graph Classification

**Wenda Li[1][\*], Kaixuan Chen[1][\*], Shunyu Liu[1], Wenjie Huang[1], Haofei Zhang[1], Mingli Song[1][†]**
**Yingjie Tian[2], Yun Su[2]**
[1]Zhejiang University, [2]State Grid Shanghai Municipal Electirc Power Company

## Abstract

In this paper, we strive to develop an interpretable GNNs' inference paradigm, termed *MSInterpreter*, which can serve as a plug-and-play scheme readily applicable to various GNNs' baselines. Unlike the most existing explanation methods, *MSInterpreter* provides a *M*essage-passing *S*election scheme (*MSScheme*) to select the critical paths for GNNs' message aggregations, which aims at reaching the self-explaination instead of post-hoc explanations. In detail, the elaborate *MSScheme* is designed to calculate weight factors of message aggregation paths by considering the vanilla structure and node embedding components, where the structure base aims at weight factors among node-induced substructures; on the other hand, the node embedding base focuses on weight factors via node embeddings obtained by one-layer GNN. Finally, we demonstrate the effectiveness of our approach on graph classification benchmarks.

## 1 Introduction

Recently, several advanced approaches (Ying et al., 2019; Luo et al., 2020; Schlichtkrull et al., 2021; Huang et al., 2022; Vu & Thai, 2020; Gui et al., 2022; Yuan et al., 2021; Schnake et al., 2021; Yuan et al., 2020; Yu & Gao, 2022) have been proposed to explain the predictions of graph neural networks (GNNs), and are divided into two categories (Yuan et al., 2023), i.e., instance and model-level explanation. The instance-level aims at critical nodes or subgraphs; on the other hand, model-level focuses on a more high-level explainability. However, most of existing GNNs' explainers are post-hoc and lack the study regarding message-passing selection for GNNs' inference.

In this paper, we aim at selecting the critical message-passing paths for GNNs' aggregations, and thus develop an interpretable GNNs' inference paradigm, i.e., *MSInterpreter*, for the task of graph classification. As shown in Figure 1, the main process is to build the *M*essage-passing *S*election scheme (*MSScheme*), and then plug it at the beginning of the existing GNNs' baselines to reach self-explainable inference. The contributions and details are summarized as follows:

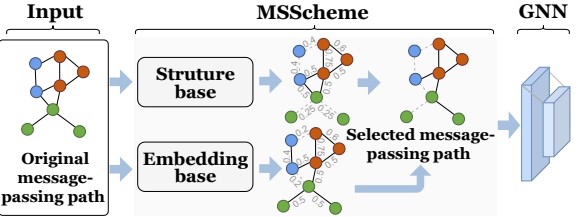

Figure 1: An illustration of the proposed *MSInterpreter* framework, i.e., plugging the *MSScheme* at the beginning of GNNs for an interpretable inference.

- We propose the *MSScheme* to calculate the weight factors among the connected nodes and then select the critical message-passing path for GNNs' aggregations.

- We plug the *MSScheme* at the beginning of an arbitrary GNN to build *MSInterpreter*, which is an end-to-end learnable framework to reach the self-explainable GNNs.

- We apply the *MSInterpreter* into graph classification task and provide the experimental analysis to support the claim that it can achieve significantly improved explanations.

---

[\*]Equal Contribution: {`lwdup,chenkx`}@`zju.edu.cn`
[†]Corresponding author: `brooksong@zju.edu.cn`

## 2 METHOD

**Notation.** A graph consisting of $n$ nodes can be represented as $G = (\boldsymbol{A}, \boldsymbol{X})$, where $\boldsymbol{A} \in \mathbb{R}^{n \times n}$ and $\boldsymbol{X} \in \mathbb{R}^{n \times d}$ represent adjacency matrix and node feature matrix respectively. For a set of labeled graphs $\mathbb{S} = \{(G_1, y_1), .., (G_N, y_N)\}$, where $y_i \in \mathbb{Y}$ denotes the label of $i$-th graph $G_i \in \mathbb{G}$. The graph classification task is to learn a function $f : \mathbb{G} \to \mathbb{Y}$ that maps graphs to the labels' set. The message-passing path is initialized as $\boldsymbol{M} = \boldsymbol{A}$. $\mathcal{N}(v)$ denotes the neighbor set of vertex $v$. To efficiently obtain the critical message aggregation path for GNNs' inference with application to graph classification, we introduce the elaborate *MSScheme* as following.

**MSScheme.** The *MSScheme* aims at the critical message-passing paths for GNNs' inference, which is the essential module to reach the self-explanation. The *MSScheme* is formulated as:

$$\boldsymbol{M} = \mathcal{M}(\boldsymbol{A}, \texttt{Mask}) \in \mathbb{R}^{n \times n} \tag{1}$$

where $\mathcal{M}$ is the message-passing selection function using mask matrix $\texttt{Mask} \in \mathbb{R}^{n \times n}$. To obtain mask matrix, we firstly compute the edge weight factors. Then, we sort these weights, and set the high-weight edges as true, and vice versa. Specifically, the edge weights are obtained by considering two components: the vanilla structure base and the node embedding base. For the vanilla structure base, we compute the edge weight between nodes $v_i$ and $v_j$ using the intersection over union (IoU) of the number of $\mathcal{N}(v_i)$ and $\mathcal{N}(v_j)$, i.e., $\texttt{W}_{str}(v_i, v_j) = \psi(\frac{num(\mathcal{N}(v_i) \bigcap \mathcal{N}(v_j))}{num(\mathcal{N}(v_i) \bigcup \mathcal{N}(v_j))})$, where $\psi$ is a norm operation in a whole graph. On the other hand, edge weight regarding node embedding is computed by $\texttt{W}_{emb}(v_i, v_j) = \varphi(h_{v_i}, h_{v_j})$, where $h_{v_i}$ and $h_{v_j}$ denote the node embeddings of $v_i$ and $v_j$ via one-layer GNN. $\varphi$ is a mapping function like line or gaussian kernel function, etc. The resulting edge factor is:

$$\texttt{W}_{com}(v_i, v_j) = \texttt{W}_{str}(v_i, v_j) + \alpha \cdot \texttt{W}_{emb}(v_i, v_j), \tag{2}$$

where $\alpha$ is a hyperparameter to balance two weight factors.

**MSInterpreter.** To build the self-explainable GNNs, we plug the proposed *MSScheme* at the beginning of the GNNs predictions. As shown in the Figure 1, we use the *MSScheme* to select the critical message aggregation path to build the interpretable GNNs' inference, i.e., *MSInterpreter*, which is beneficial to build an end-to-end framework while training GNNs for the task of graph classification.

## 3 EXPERIMENT

We compare *MSInterpreter* with four popular explainable methods PGExplainer (Luo et al., 2020), GNNExplainer (Ying et al., 2019) ,SubgraphX (Yuan et al., 2021), and GStarX (Zhang et al., 2022). In Table 1, we list these methods' accuracy, recall, and F1 score, and our method achieves the competitive performances. More experimental details are provided in Appendix A.

Table 1: Evaluation of several explainable methods on two graph classification datasets (with annotated explanatory edges) using a three-layer GIN architecture.

| Dataset | BA-2MOTIFS | | | MUTAG$_0$ | | |
|---|---|---|---|---|---|---|
| | Acc. | Rec. | $F_1$ | Acc. | Rec. | $F_1$ |
| GNN-Exp. | $49.21_{\pm 0.39}$ | $52.65_{\pm 1.91}$ | $30.85_{\pm 0.96}$ | $50.48_{\pm 1.82}$ | $59.22_{\pm 0.70}$ | $48.00_{\pm 0.80}$ |
| PGE-Exp. | $36.05_{\pm 0.11}$ | $80.23_{\pm 0.57}$ | $35.13_{\pm 0.21}$ | $56.27_{\pm 0.59}$ | $83.60_{\pm 0.89}$ | $59.44_{\pm 0.56}$ |
| SubgraphX | $\mathbf{71.15}_{\pm 1.00}$ | $52.08_{\pm 0.82}$ | $40.42_{\pm 0.18}$ | $69.05_{\pm 0.91}$ | $38.35_{\pm 2.90}$ | $48.62_{\pm 2.68}$ |
| GStarX | $61.85_{\pm 0.01}$ | $86.86_{\pm 1.67}$ | $49.56_{\pm 0.45}$ | $46.96_{\pm 0.01}$ | $74.62_{\pm 1.85}$ | $58.88_{\pm 0.24}$ |
| MSInterpreter | $66.34_{\pm 0.37}$ | $\mathbf{90.78}_{\pm 0.87}$ | $\mathbf{53.75}_{\pm 0.53}$ | $\mathbf{79.31}_{\pm 0.95}$ | $\mathbf{88.69}_{\pm 1.63}$ | $\mathbf{76.12}_{\pm 1.04}$ |

## 4 CONCLUSIONS

In this paper, we introduce an novel framework to explain the GNNs' inference with application to graph classification. Firstly, we build a scheme *MSScheme* to analyze the weight factors of message-passing paths, which is essential to obtain the crucial message aggregations for GNNs' inference. Then, we plug the *MSScheme* at the begin of the GNNs' predictions to build an end-to-end interpretable paradigm, i.e., *MSInterpreter*, which aims at reaching the self-explaination GNNs for graph classification. Finally, we perform our proposed method in graph classification to demonstrate its superior performance.

## ACKNOWLEDGEMENTS

This work is supported by the Science and Technology Project of SGCC: Hybrid enhancement intelligence with human-AI coordination and its application in reliability analysis of regional power system (5700-202217190A-1-1-ZN).

## URM STATEMENT

The authors acknowledge that at least one key author of this work meets the URM criteria of ICLR 2023 Tiny Papers Track. Author Wenda Li meets the URM criteria of ICLR 2023 Tiny Papers Track.

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

## A APPENDIX

**Dataset.** We compare two datasets with edge interpretation labels: $MUTAG_0$ and BA2Motifs. The statistics for the BA2Motif (Luo et al., 2020) and $MUTAG_0$ (Tan et al., 2022) datasets are shown in Table 2.

- $MUTAG_0$ is a molecular dataset for graph classification tasks and consists of 4,337 molecular graphs. In this dataset, nodes represent different atoms, and edges represent chemical bonds. Each graph is assigned to one of the two categories according to its mutagenic effects (Riesen & Bunke, 2008). We observe that carbon rings are present in both mutagenic and non-mutagenic graphs, but the carbon rings with the chemical groups $NH_2$ or $NO_2$ are mutagenic. Therefore, the carbon ring can be considered a shared base map, and the two groups $NH_2$ and $NO_2$ are the base sequence of the mutagenic map. For the non-mutagenic graphs, there is no explicit base sequence. For the mutagenic graphs, there are true edge masks that mark the edges of the mutagenic motifs.
- BA2Motifs is a synthetic graph motif detection dataset. BA2Motifs contains Barabasi-Albert (BA) base graphs of size 20, and each graph has five node patterns. The node features are 10-dimensional all-one vectors. Patterns can be house-like structures or cycles. The graphs are divided into two categories based on the topics they contain. Therefore the interpretation of this dataset is the edges that make up the two patterns.

**Experiments Settings.** We use three-layer GIN as the backbone network and set the hidden dimensions as 128. We divide the dataset into three random groups (80%/10%/10%) as training, validation, and test sets.

For the BA2Motifs dataset, we compute the three metrics only for correctly predicted data. For the $MUTAG_0$ dataset, we add one more restriction that the prediction category is mutagenic since only mutagenic data have interpreted edges. We uniformly set the batch size to 64 and the number of epochs to 100. For our method, we keep the sparsity at 0.5, and the hyperparameter $\alpha$ is set to 0.5.

Our implementation is based on Python 3.8.15, PyTorch 1.12.0, PyTorchGeometric 2.2.0 (Fey & Lenssen, 2019), and DIG (Liu et al., 2021). We adopt the GNN implementation and the provided implementation of the baseline explainer from the DIG library.

**Graph Classification.** We provide the accuracy of graph classification in table 3. Compared to the four post-hoc explainers, our method keeps the accuracy no less than the accuracy of backbone.

Table 2: Dataset Statistics.

| Dataset | Graphs | Edges(avg.) | Nodes(avg.) | Classes |
|---------|--------|-------------|-------------|---------|
| $MUTAG_0$ | 2301 | 32.54 | 31.74 | 2 |
| BA2Motifs | 1000 | 25.00 | 51.39 | 2 |

Table 3: Accuracy of graph classification for various interpretation methods

| | GIN | GNN-Exp. | PGE-Exp. | SubgraphX | GSTtarX | MSInterpreter |
|---|-----|----------|----------|-----------|---------|---------------|
| $MUTAG_0$ | 0.995 | 0.826 | 0.995 | 0.904 | 0.934 | 1.0 |
| BA2Motifs | 1.0 | 0.993 | 1.0 | 0.931 | 0.964 | 1.0 |

**Discussion.** The developments of graph neural networks (GNNs) have revolutionized the domains non-Euclidean space (Jing et al., 2022; Chen et al., 2023). Especially, graph representation learning and other graph symmetries with application to various graph downstream tasks are well studied, e.g., vision tasks (Jing et al., 2021; Zhang et al., 2023), cell clustering (Alghamdi et al., 2021; Li et al., 2022), chemical prediction (Tavakoli et al., 2022; Zhong et al., 2022; Chen et al., 2022b), reinforcement learning (Liu et al., 2022) and power system (Boyaci et al., 2021; Chen et al., 2022a; Han et al., 2022). However, the predictions of these GNN baselines lack explainability, i.e., their

inferences for handling graph-structure data are still treated as black boxes, which limits their applications in some crucial areas.

Recently, several advanced approaches (Ying et al., 2019; Luo et al., 2020; Schlichtkrull et al., 2021; Huang et al., 2022; Vu & Thai, 2020; Gui et al., 2022; Yuan et al., 2021; Schnake et al., 2021; Yuan et al., 2020; Yu & Gao, 2022) have been proposed to explain the predictions of graph neural networks (GNNs), and are divided into two categories (Yuan et al., 2023), i.e., instance and model-level explanation. The instance-level aims at critical nodes or subgraphs; on the other hand, model-level focuses on a more high-level explainability. *However, most of existing GNNs' explainers are post-hoc and lack the study regarding message-passing selection for GNNs' inference.*

