# OpenReview forum: "Message-passing Selection: Towards Interpretable GNNs for Graph Classification"
_ICLR.cc/2023/TinyPapers — Submitted to Tiny Papers @ ICLR 2023_

### Official Review · Reviewer_4497 · 2023-03-29

**Confidence:** 3

**Summary Of Contributions:**

The authors present MSInterpreter, a novel plug-and-play scheme for interpretable inference in Graph Neural Networks (GNNs). The proposed Message-passing Selection scheme (MSScheme) identifies critical paths for message aggregation, focusing on self-explanation. Experiments demonstrate its effectiveness on graph classification benchmarks, offering a promising approach to enhance GNN interpretability.

**Rating:**

Clear, Correct, and Reproducible (CCR): a submission which meets the reviewing criteria

**Strengths And Weaknesses:**

Strengths:

1. MSInterpreter's compatibility with various GNN baselines allows for easy integration and broader applicability in the field.

2. By emphasizing self-explanation instead of post-hoc explanations, MSInterpreter enables more transparent and intuitive understanding of GNN inferences.

3. The paper showcases the effectiveness of the proposed approach through experiments on graph classification benchmarks, strengthening the credibility of the method.

**Suggested Changes:**

None

---

### Official Review · Reviewer_n8KF · 2023-03-30

**Confidence:** 4

**Summary Of Contributions:**

The paper proposes an interpretable graph neural network (GNN) inference paradigm called MSInterpreter, which can be applied to various GNN baselines. MSInterpreter provides a Message-passing Selection scheme (MSScheme) to select the critical paths for GNNs' message aggregations, which aims to reach self-explanation instead of post-hoc explanations.

**Rating:**

Clear, Correct, and Reproducible (CCR): a submission which meets the reviewing criteria

**Strengths And Weaknesses:**

## Strengths

- The paper addresses an important and relevant problem of interpretability in graph neural networks (GNNs), which are increasingly used in various applications.

- The proposed MSInterpreter framework is innovative and provides an interpretable approach for GNNs' inference by selecting critical message-passing paths.

- The paper provides a clear and detailed description of the proposed method, including the formulation of the MSScheme and the experimental setup.

## Weaknesses

- For the structural similarity, the authors compute the edge weight between two nodes by taking into account just the number of common neighbors of the two nodes. However, in many tasks, two nodes can be structurally similar even if they do not have any common neighbor, For example, assume two nodes with start neighborhoods. Even if they do not share any common neighbor, they exhibit the same structure, and the current method doesn't take into account that.



**Suggested Changes:**

The paper could benefit from a more detailed discussion of the limitations and future directions of the proposed method. Also, I would suggest the authors take into account the structural similarity of the nodes and not rely just in their common neighbors.

---

### Official Review · Reviewer_kqDz · 2023-04-02

**Confidence:** 4

**Summary Of Contributions:**

This paper focus on interpretable GNN. The paper proposes  MSInterpreter, an interpretable GNN paradigm that selects critical message aggregation paths using structural and node embedding components, proving effective on graph classification benchmarks.

**Rating:**

High Impact (HI): a submission which meets the reviewing criteria and is predicted to make an impact on the field

**Strengths And Weaknesses:**


**Strengths**
1. The paper is well-structured, with a clear and coherent presentation that makes it easy for readers to follow.
2. The focus on interpretability at the graph level is significant, as it addresses a crucial aspect of understanding complex graph-based models.
3. The proposed method is both intuitive and logical, as it efficiently selects the critical paths for GNNs' message aggregation, which is essential for improving the interpretability of the model.
4. The experimental result shows the effectiveness of the proposed method.



**Weaknesses**

I don’t see any significant weakness in this paper.


**Suggested Changes:**

N/A

---

### Meta-Review · Area_Chair_88pq · 2023-04-04

**Recommendation:** Invite to present
**Confidence:** 4

**Metareview:**

All reviewers, as well as myself, agreed that this paper is Clear, Correct, and Reproducible (CCR). It is also potentially of high interest to the readership of this venue.

**Summary:**

This well-written paper proposes MSInterpreter, a novel GNN interpreter that select message-passing paths.

**Reason For Not Giving A Higher Recommendation:**

URM statement is missing.

**Reason For Not Giving A Lower Recommendation:**

The proposed method is innovative and effective. The paper is well written.

---

### Decision · Program_Chairs · 2023-04-10

**Decision:**

Invite to present

**Comment:**

Please add URM Statement section to camera-ready.

---

> ### Author Response · Authors · 2023-06-01
> **Archivization of this paper**
>
> We wish to opt in for archival of our paper